# Comparative Activation Process of Pb, Cd and Tl Using Chelating Agents from Contaminated Red Soils

**DOI:** 10.3390/ijerph17020497

**Published:** 2020-01-13

**Authors:** Lirong Liu, Dinggui Luo, Guangchao Yao, Xuexia Huang, Lezhang Wei, Yu Liu, Qihang Wu, Xiaotao Mai, Guowei Liu, Tangfu Xiao

**Affiliations:** 1School of Environmental Science and Engineering, Guangzhou University, Guangzhou 510006, China; leonliutiming@e.gzhu.edu.cn (L.L.); yaoguangchao@e.gzhu.edu.cn (G.Y.); huangxuexia66@163.com (X.H.); wlz2016@gzhu.edu.cn (L.W.); liuyu@gzhu.edu.cn (Y.L.); maixiaotao@e.gzhu.edu.cn (X.M.); 2111704012@e.gzhu.edu.cn (G.L.); tfxiao@gzhu.edu.cn (T.X.); 2Linköping University—Guangzhou University Research Center on Urban Sustainable Development, Guangzhou University, Guangzhou 510006, China; 3Key Laboratory for Water Quality and Conservation of the Pearl River Delta, Ministry of Education, Guangzhou University, Guangzhou 510006, China; wuqihang@gzhu.edu.cn; 4Guangdong Provincial Key Laboratory of Radionuclides Pollution Control and Resources, Guangzhou University, Guangzhou 510006, China

**Keywords:** heavy metal, thallium, chelating agents, phytoextraction, activation, red soil

## Abstract

Adding chelating agents is a critical technique of heavy metal activation for enhancing phytoextraction through the formation of soluble metal complexes which will be more readily available for extraction. The preliminary, dynamic, equilibrium activation experiments and speciation analysis of Pb, Cd and Tl in contaminated red soils were used to select six chelates with relatively good activation performance from nine chelates, and the effects of dosage and pH on the heavy metals activation were studied systematically. Results showed that the activation of Pb, Cd and Tl by chelates reached equilibrium within 2 h, and the activation process showed three stages. Under neutral conditions, chelates had better activation performance on Pb- and Cd-contaminated soils. Except for S,S-ethylenediamine disuccinic acid (S,S-EDDS) and citric acid (CA), the maximum equilibrium activation effect (MEAE) of ethylenediaminetetraacetic acid (EDTA), N,N-bis (carboxymethyl) glutamic acid (GLDA), diethylenetriaminepentaacetic acid (DTPA) and aminotriacetic acid (NTA) was over 81%. The MEAE of Tl-contaminated soil was less than 15%. The decreasing order of the dosage of chelating agents corresponding to MEAE for three types of contaminated soils was Pb-, Cd- and Tl-contaminated soil, relating to the forms of heavy metals, the stability constants of metal–chelates and the activation of non-target elements Fe in red soil. Under acidic conditions, the activation efficiencies of chelates decreased to differing degrees in Pb- and Cd-contaminated soils, whereas the activation efficiencies of chelating agents in Tl-contaminated soils were slightly enhanced.

## 1. Introduction

As a consequence of ongoing rapid economic development and expanding urbanization, increasingly large amounts of waste are being discharged into the soil. Among these wastes, heavy-metal pollutants, which are well known as high bioaccumulation, recalcitrant, and toxic, are particularly serious [1,2,3]. Heavy metals can enter into the human body via the food chain, posing serious threats to the environment and human health. Therefore, it is essentially important to remove and control the heavy metals from contaminated soil for its remediation, which has become an urgent problem to be solved around the world [4,5].

Compared with the conventional soil remediation methods, phytoremediation is regarded as an environmentally-friendly, low-cost, highly efficient method; most importantly, it causes less disturbance to the environment [6,7]. Phytoextraction, as one of the commonly used phytoremediation techniques, is based on the principle that toxic and harmful substances are transported from contaminated soil to plant shoots via absorption from the roots of the plant and then removed by harvesting the plants [8,9,10,11,12]. In the Cantabrian Mountains of southern Spain, areas contaminated by heavy metals from slag have been remediated by means of phytoextraction [13]. As a means of potentially enhancing the efficacy of phytoextraction, chelation-enhanced phytoextraction technology, which promotes the activation of heavy metals from the surface of soil particles into the soil by adding chelating agents, has been widely recognized by the majority of scientific research workers [14,15].

Commonly used chelating agents can be divided into three categories. The first is aminopolycarboxylic acids (APCAs), such as ethylenediaminetetraacetic acid (EDTA), diethylenetriaminepentaacetic acid (DTPA), S,S-ethylenediamine disuccinic acid (S,S-EDDS), aminotriacetic acid (NTA), and l-glutamic acid, N,N-bis (carboxymethyl) glutamic acid (GLDA) [16,17]. These chelating agents have a strong ability to complex Pb and Cd, and consequently promote plant uptake [18,19]. The second is natural low molecular weight organic acids (NLMWOAs), including CA (citric acid), OA (oxalic acid), and AA (acetic acid) [20]. The third is surfactants such as RH (rhamnose) [21]. To date, it has been established that numerous chelating agents exhibit different degrees of activation for lead (Pb), cadmium (Cd), copper (Cu), and zinc (Zn), and other heavy metals [22,23,24].

Red soil is a typical type of soil in the south of China with high iron oxide content and low organic matter content, which has a great impact on chelate activation for heavy metals from contaminated soil. Red soil areas are also one of the main distribution areas of non-ferrous metal mines, where heavy metal pollution (such as Pb and Cd) caused by mining is widespread. Moreover, acid rain is a typical climatic feature in the region of south China. At present, there are few systematic studies on the activation characteristics of heavy metal pollution in red soil and the effect of acid rain on activation [25]. In addition, thallium (Tl)-contaminated soil, caused by the excavation activities of non-ferrous metal mines, is also a typical pollution type. Although Tl is a very toxic element, there is relatively rare research on chelating-enhanced phytoextraction for Tl [26,27].

In this study, the preliminary, dynamic and equilibrium activation experiments were conducted to screen typical chelating agents (three types of nine chelating agents EDTA, EDDS, NTA, DTPA, GLDA, OA, CA, AA, and RH), and the effects of chelating agents dosage and pH conditions on the activation of Pb-, Cd-, and Tl-contaminated red soils was systematically investigated. Furthermore, combining with heavy metal speciation analysis, the mechanism of heavy metal activation was investigated, which provided potential application for selecting appropriate chelating agents for chelate-assisted phytoremediation of heavy metals in red soil.

## 2. Materials and Methods

### 2.1. Materials

#### 2.1.1. Soil Preparation

The soil samples used in this study were collected from the forest top red soil (0–20 cm, red soil) near a pyrite area (22°59′25.5″ N, 112°00′40.5″ E) in Yunfu, Guangdong Province, China. Soils were dried under natural conditions. The dried soils were screened to remove large pieces of plant debris and stones, crushed, sieved through a 4-mm nylon sieve, boxed, and preserved for further use. The physicochemical properties of the soil sample determined in our previous study [28] are listed in Table 1.

#### 2.1.2. Preparation of Contaminated Soil

In order to simulate heavy metal pollution in red soil, we artificially added CdCl_2_·2.5H_2_O, Pb(NO_3_)_2_, and TlNO_3_ to the prepared soil to further prepare Pb-, Cd-, and Tl-contaminated soils with 1000, 10 and 10 mg/kg, respectively. The contaminated soils underwent cycles of wetting (70% field capacity) and drying (air-drying) for 60 days to obtain a geochemical equilibration.

#### 2.1.3. Experimental Reagents

EDTA, DTPA, EDDS, NTA, OA, CA, AA, and RH were all of analytical grade and were purchased from Zhiyuan Chemical Reagent Co., Ltd. (Tianjin, China), whereas GLDA was purchased from TCI Chemical Co., Ltd. (Shanghai, China). Solutions of all nine chelating agents were prepared separately at a concentration of 7.5 mmol/L.

### 2.2. Experimental Design and Operation

#### 2.2.1. Preliminary and Dynamic Activation Experiment

A 2 h preliminary activation experiment was conducted for each of the three single-contaminated soils (Pb-, Cd-, and Tl-contaminated soils). Then. according to the experimental results, the chelating agent with higher activation efficiency was selected for dynamic activation experiment.

For dynamic activation experiments, we set a series of time points for the observation of the activation effect on heavy metals with the chelating agents selected in the previous step, in order to study the characteristics of the activation process and determine the sufficient time when the activation reaches equilibrium, and the corresponding maximum activation efficiency.

The experimental procedures of preliminary and dynamic activation experiments were conducted as reported by the previous study [29,30,31]: the prepared contaminated soil (1.0000 ± 0.0001 g) that had been sieved through a 100 mesh was placed in a 50-mL centrifuge tube in which was added 8.0 mL of each chelating agent solution and 1 mL of 0.01 mg/L NaNO_3_ solution to maintain ionic strength. The water to soil ratio was maintained at 10:1, with the pH under 6.5 by HNO_3_ and NaOH. After shaking in a 25 °C water bath for 1, 5, 15, 30, 60, 120, 240, and 480 min (preliminary activation experiment, 120 min), the suspension was centrifuged at 3500× *g* for 5 min and the resulting supernatant was filtered through a 0.45-micron membrane filter. The concentrations of Cd, Pb, and Tl in the filtrate were determined by atomic absorption spectrophotometry to calculate the activation efficiency [32]. Each group of experiments was carried out three times in parallel, a total of 174 groups (27 and 144 groups for static and dynamic experiment, respectively, and including 3 blank).

Activation efficiency f was determined using the following equation:$$f=\frac{\left(C_{M}-C_{0}\right)V}{M_{S}}\times 100\%$$
where $C_{M}$ was the content of heavy metal in solution after the activation of chelating agent, $C_{0}$ was the content of heavy metal in solution without chelating agent, V was volume of solution, and $M_{s}$ was the mass of heavy metals in the soil.

#### 2.2.2. Influence of Chelating Agent Dosage and Solution pH on the Activation Effect of Heavy Metals

To assess the influences of the dosage of chelating agents and pH environment conditions on the activation effect, we designed different dosages of chelating agents to the red soils contaminated with Pb, Cd, and Tl, fixed the activation time when activation reaches equilibrium (known as the equilibrium activation experiment), and observed the equilibrium activation effects of heavy metals under two pH conditions (4 and 6.5, representing acid rain and non-acid rain, respectively). The specific procedure was as follows.

Different volumes (1.0, 2.0, 3.0, 4.0, 5.0, 6.0, 7.0, and 8.0 mL) of the typical chelating agent solutions selected from the previous experiment were added into the centrifuge tubes, and the solution pH was adjusted to either 4 or 6.5. The remainder of the procedure was the same as that described in Section 2.2.1. Each group of experiments was performed three times in parallel for a total of 288 groups. The activated soil was filtered and dried and left for sequential heavy metal speciation analysis.

#### 2.2.3. Determination of Heavy Metal Speciations before and after Activation

For each of the three types of contaminated soils, a typical chelating agent was selected to examine the speciation changes of heavy metals before and after activation in the experiment. The specific procedure used was as follows.

We used a modified BCR three-step sequential extraction method [33] to determine the speciation of heavy metals before and after soil activation using the following four steps: (1) To examine the exchangeable fraction, we weighed 0.5 g of soil into a 50-mL centrifuge tube in which was added 20 mL 0.11 M acetic acid (pH = 2.8). The tubes were then shaken in a water bath at 25 °C for 16 h, followed by centrifugation at 3500× *g* for 15 min. The resulting supernatant was taken out, and subsequently added to 10 mL of deionized water, followed by shaking for 15 min to ensure that there was no residual sticky soil, and then the centrifugation was repeated and the supernatant collected. After repeating this procedure, the resulting supernatant was made up to a final volume of 10 mL. (2) To examine the reducible fraction, we took the residue from Step 1 and added 20 mL of 0.5 M hydroxylamine hydrochloride (pH = 1.5). After shaking and centrifugation, the subsequent procedures were the same as those described in Step 1. (3) To examine the oxidizable fraction, we took the residue from Step 2 and added 5 mL of hydrogen peroxide (pH = 2.2), shaken at 25 °C for 1 h, opened the lid and steamed at 85 °C. Thereafter, we added 5 mL of hydrogen peroxide, tightened the lid, and heated for 1 h in a water bath at 25 °C, and then opened the lid to dryness in 85 °C, cooled to room temperature, and added 25 mL of 1 M ammonium acetate (adjusted to pH = 2.0 with nitric acid). (4) The residual fraction was calculated as the difference between the pseudo-total concentration of heavy metal (Pb, Cd or Tl) extracted with aqua regia digestion and the sum of heavy metal concentrations in the three fractions described above.

### 2.3. Statistical Analysis and Processing Methods

The data were analyzed and plotted using the Origin software package. Data are presented as the means ± standard deviation (mean values ± SD). All data means were compared using an analysis of variance (*T*-test, *p* < 0.05).

## 3. Results

### 3.1. Screening of Chelating Agents and Dynamic Activation Experiment

The results of the preliminary activation experiments, and activation efficiency of the nine chelating agents EDTA, EDDS, NTA, GLDA, DTPA, OA, AA, CA, and RH on the soils contaminated with Pb, Cd, and Tl are shown in Figure 1.

As can be seen from Figure 1a for Pb-contaminated soils, the descending order of chelates for Pb activation ability was EDTA, DTPA, NTA, GLDA, EDDS, CA, OA, AA and RH. The APCA category of chelating agents showed a good activation effect, with activation efficiencies in the range of 55.51% to 89.16%. The low molecular weight organic acids, except CA with 11.52% activation efficiency, OA, AA and surfactant RH showed low activation efficiencies, less than 3%. For Cd in Figure 1b, the descending order for Cd activation ability was NTA, GLDA, EDTA, EDDS, DTPA, CA, RH, AA and OA. Similarly, the APCA showed a good activation effect, with activation efficiencies in the range of 64.61% to 94.40%. The low molecule weight organic acids, except for CA with 47.41% activation efficiency, OA, AA and surfactant RH also had a low activation property and less than 5%. For Tl-contaminated soil in Figure 1c, the activation efficiencies of nine chelating agents is low, less than 5%, especially of OA, AA, CA, RH which were particularly low, less than 1%.

Based on the experimental results, the six chelating agents (EDTA, EDDS, NTA, DTPA, GLDA, and CA) with greater activation efficiency were selected for further study.

The dynamic experimental results with six chelating agents to three types of contaminated soil are shown in Figure 2. For each of the soils contaminated with Pb, Cd, and Tl, the activation efficiency of the six typical chelating agents showed similar characteristics over the time, which can be divided into three stages: fast, slow, and equilibrium, and reached activation equilibrium within 2 h.

### 3.2. The Effect of Chelating Agent Dosage and Solution pH on Heavy Metals Activation

The relationship between the dosage of chelating agent and the equilibrium activation efficiency (EAE) were investigated at different pH levels by equilibrium activation experiment. Figure 3, Figure 4 and Figure 5 show the relation curves between the dosage of chelating agents and the EAE of Pb-, Cd- and Tl-contaminated soils under two pH (6.5, 4) conditions.

(1) Under the condition of neutral pH (6.5): for both Pb- and Cd-contaminated soils (Figure 3 and Figure 4), the change of EAE of the six chelating agents is divided into three stages (fast, slow and gradually tend to stable) with the increase of the chelating agent dosage, but there were some differences in the inflection points at which the maximum equilibrium activation efficiency (MEAE) was obtained.

For the Pb-contaminated soil (Figure 3), descending order of MEAE of the six chelating agents at inflection point was EDTA (90.91%) > NTA (89.62%) > DTPA (89.13%) > GLDA (87.13%) > EDDS (76.38%) > CA (50.03%), whereas the corresponding chelating agents dosage was EDTA (3.11), NTA, DTPA, and GLDA (all 4.66), EDDS (9.32), and CA (10.88). With the exception for relatively low EDDS (76.38%) and CA (50.03%), the MEAE of the other four chelating agents (EDTA, NTA, DTPA, GLDA) are all above 87%, and the corresponding dosage of chelating agents varies from 3 to 11.

For the Cd-contaminated soil (Figure 4), the descending order of MEAE of the six chelating agents at the inflection point was EDTA (86.96%) > DTPA (85.39%) > GLDA (83.69%) > NTA (80.98%) > CA (76.11%) > EDDS (69.07%), whereas the corresponding chelating agents dosage was EDTA (56.21 unit), DTPA, GLDA and NTA (all 421.54 unit), CA and EDDS (both 505.85 unit). Similar to Pb-contaminated soils, the MEAE of CA (76.11%) and EDDS (69.07%) were relatively low, and that of the other four chelators (EDTA, DTPA, GLDA, NTA) were between 87% and 81%. The corresponding dosage of chelating agents ranged from 56 to 506 unit, which was much higher than that of Pb-contaminated soil.

The EAE in Tl-contaminated soil differed significantly from that in the Pb- and Cd-contaminated soils, as the three-stage change pattern was not detected (Figure 5). It is considered that the activation of heavy metals in this soil was still in the first or second stage. However, it is unnecessary to continue the observations due to the extremely large dosage of chelating agents. The descending order of the MEAE under the conditions of this experiment was as follows: EDTA (15.48%) > CA (9.55%) > DTPA (6.05%) > EDDS (4.57%) > GLDA (3.53%) > NTA (3.31%), all less than 15%. For all kinds of chelating agents, the dosage of chelating agent is much greater than that of Cd.

(2) Changes of activation characteristics of chelating agents in acidic pH environment.

For Pb-contaminated soil (Figure 3), the activation efficiencies of the six chelating agents under neutral conditions were all better than those in an acidic environment. Moreover, the difference of EAE under two pH conditions is different for several chelating agents. The maximum differences of EAE for EDTA, NTA, and DTPA are less than 5.46%, and tended to decrease with an increasing dosage of chelating agent. The most significant differences were observed among EDDS and CA, for which the difference of EAE was 35.93% and 54.13%, and the difference increased with an increase of chelating agent dosage. Values for GLDA were intermediate between those of EDDS and CA, with a maximum difference in EAE of 47.21%, and the difference decreased significantly with an increase in the dosage of chelating agent.

For Cd-contaminated soil (Figure 4), with the exceptions of NTA and DTPA, which showed slightly better activation under acidic conditions than in the neutral environment (the maximum difference in EAE was 1.31 to 4.45%), the remaining four chelating agents all performed better under neutral pH conditions, among which EDDS and CA showed the most significant differences, with 48.87% and 52.82% respectively. Moreover, the differences increased with an increase in the dosage of chelating agent. Secondly, the maximum difference in GLDA was 48.50% and decreased with the addition of chelating agents. EDTA showed the smallest difference in EAE, with a maximum difference of less than 0.95%.

For the Tl-contaminated soil (Figure 5), the activation effects of the six chelating agents were slightly better under acidic conditions than in a neutral environment, although the maximum difference in EAE was no more than 10%.

### 3.3. Speciation Characteristics of Heavy Metals before and after Activation

In the speciation analysis experiment (Figure 6), we selected three samples, namely, EDTA-activated Tl-contaminated soil, EDDS-activated Cd-contaminated soil, and CA-activated Pb-contaminated soil, in order to examine the changes in the speciation characteristics of heavy metals before and after activation. The results are shown in Figure 6, in which the form of heavy metals is represented by the horizontal coordinates, arranged in the order of activation from easy to difficult (exchangeable, reducible, oxidizable, and residual), and the contents of heavy metals in different forms are represented by the vertical coordinates.

(1) For Pb-contaminated soil, the exchangeable, reducible, oxidizable, and residual fractions of the heavy metals before activation were 449.77, 445.50, 52.62, and 52.12 mg/kg, accounting for 45%, 45%, 5%, and 5%, respectively. Exchangeable and reducible fractions are the dominant forms, with a percentage of 90%. After activation, the values were 149.60, 239.47, 38.86, and 42.79 mg/kg, respectively. In contrast to the oxidizable and residual fractions, which decreased slightly, the exchangeable and reducible fractions decreased by 300.17 and 206.03 mg/kg (66.74%, 46.25%), respectively, and by 506.20 mg/kg in total for both fractions, which was similar to the activation efficiency of CA in the activation experiment. Further calculations showed that the activation by CA was relatively low, as 389.06 mg/kg of the exchangeable and reducible fractions (accounting for 43.46%) remained unactivated, which means a strong potential for activation.

(2) In the Cd-contaminated soils, the exchangeable, reducible, oxidizable, and residual fractions of the heavy metals before activation were 6.70, 1.66, 0.75, and 0.89 mg/kg, accounting for 67%, 17%, 7%, and 9%, respectively. Exchangeable and reducible fractions are also the dominant forms, with a percentage of 84%. After activation, Cd in all four speciations decreased in different degrees, with a total reduction of 7.94 mg/kg, which is essentially consistent with the activation efficiency of 79.04% mentioned previously (the MEAE of Cd with EDDS under neutral pH). The number of exchangeable and reducible fractions decreased by 82%, and the remaining 18%, indicating that there was small potential for further activation.

(3) For Tl-contaminated soils, values for the exchangeable, reducible, oxidizable, and residual fractions of the heavy metal before activation were 2.1383, 2.0717, 0.8109, and 4.9491 mg/kg, accounting for 21%, 21%, 8%, and 49%, respectively. Reducible fraction is the most dominant form, followed by exchangeable fraction, reducible fraction and oxidizable fraction. After chelate activation, the values changed to 1.0347, 1.9360, 0.7408, and 4.4854 mg/kg, respectively. The total reduction was 1.502 mg/kg, which is essentially the same as the EDTA activation result of 1.498 mg/kg obtained in the activation experiment. After activation, only the exchangeable fraction showed a substantial reduction, and there was still 1.0347 mg/kg (accounting for 48.39%) of the exchangeable fraction which remained unactivated, indicating the potential for further activation.

## 4. Discussion

In general, chelating agents activate the heavy metal through the formation of soluble metal-chelates, which would improve the release of heavy metal on the soil surface, positively impacting the phytoextraction. However, the activation efficiency of chelating agents comprehensively depends on the structure of chelating agents, the types and speciation of heavy metals and soil condition (such as pH). The result of the preliminary experiment demonstrated that the activation effect of DTPA, EDTA, EDDS, GLDA, NTA, and CA was relatively high, while the activation of low molecular weight organic acids OA, AA and surfactant RH was particularly low. The activation efficiency of chelating agents is related to the coordination atoms that they provide [34]. On the basis of the chemical structure of chelating agents, DTPA, EDTA, EDDS, GLDA, NTA, and CA can form 8, 6, 6, 5, 4 and 4 coordinate bonds, respectively, which enable these agents to capture heavy metal ions on the surface of soil particles and promote the formation of stable complexes, thus enhancing the activation efficiency. Oxalic acid (OA) and acetic acid (AA) provide only 2 and 1 coordination atoms, respectively, and generate insoluble lead oxalate and cadmium oxalate; therefore, the activation effect of these chelating agents tends to be poor, and even could not be measured by atomic absorption spectrophotometry in the present work. In addition, the surfactant RH, a bipolar molecule, composed of a hydrophilic polar head and a hydrophobic non-polar tail, also has a low activation effect. When rhamnose interacts with heavy metals, the hydrophobic end binds to the contaminant, whereas the hydrophilic end protrudes outward, with only 2 binding bonds reacting with heavy metals. Thus, the activation effect is relatively poor [35].

The activation effect of chelating agents can also be explained by the chelate stability constant (lgK) for the interaction between chelating agents and heavy metals. The higher value of the constant, the better is the activation effect [36,37,38]; for example, the values for Cd-EDTA and Cd-GLDA (16.5 and 10.31, respectively) are higher than that for Cd-CA (7.9). Furthermore, the activation effect of chelating agents is also closely related to the type of heavy metal [39]. We, accordingly, noted the significant differences in the activation effects for the three heavy metals in the present study.

Based on the dynamic experimental results, for each soil contaminated with Pb, Cd and Tl, the activation efficiency of the six typical chelating agents show changes in fast, slow and tend to equilibrium over time. We used quasi-first-order and quasi-second-order kinetic models to simulate the activation process of heavy metals by chelating agents [40,41], the results showed that pseudo-second-order kinetic model fits quite well to the activation of Pb, Cd, and Tl for the six selected chelating agents with the goodness of fit more than 0.9878, thereby reflecting the chemical activation characteristics of the chelating agents [42].

The adsorption of heavy metals on soil, particularly red soil, can be classified into non-specific and specific adsorption [43]. Non-specifically adsorbed heavy metal ions have better activity and migration ability, and interact readily with chelating agents, resulting in easy desorption from the surface of soil particles. As to specific adsorption, adsorbed heavy metals tend to have poor activity and migration ability and they can only be desorbed from the soil by stronger chemical complexation or chelation. In the process of soil heavy metal activation, non-specifically adsorbed heavy metal ions are rapidly desorbed into solution at an early stage of activation, and there is an early rapid increase in the concentration of heavy metals in solution. With increasing time, there is a gradual transition to the desorption of specifically adsorbed heavy metal ions, and so the activation process becomes slower and eventually reaches a dynamic equilibrium.

The EAE of the six chelating agents showed a pronounced three-stage pattern of fast, slow, and relatively stable with an increase of the chelating agents dosage of Pb- and Cd-contaminated soils under neutral pH conditions. Further statistical analysis shows the EAE of the six chelating agents approximately obeyed the law of logarithmic change with the increase of the chelating agents dosage, coefficients of determination ranging from 0.693 to 0.984. In this study, Pb- and Cd-contaminated soils have better activation effect, which is closely related to the existing forms of Pb and Cd, and most of them exist in exchangeable and reducible fractions, and these results were consistent with other researchers [44,45,46].

The EAE in Tl-contaminated soil differed significantly from that in the Pb- and Cd-contaminated soils, and the three-stage change pattern was incomplete. Under the present experimental conditions, the MEAE of Tl is below 15%, which is mainly related to the existential speciation of Tl, the stability constant of complexation reaction and the activation of non-target element Fe in red soil. In red soil, Tl mainly existed as residual fraction (based on the results of heavy metal speciation analysis), which is one of the important reasons for the low activation ability. Second, taking EDTA as an example, the complexation stability constant (lgK) of Tl is much smaller than that of Pb, Cd (Pb, Cd and Tl, respectively, 18.0, 16.5, 2.3) [36,47]. In addition, because the complexation stability constant of EDTA and Fe is higher (lgK = 25.1), and the stability constant of the target element is much lower than Fe, the activation of the target element will be greatly weakened. Some researches showed that Fe can occupy the site of ligand, which would weaken the chelated activation to the heavy metal [48,49,50,51]. The activation efficiency of the target element can only be improved by adding larger dosages of chelating agent; thus, with the dosage of chelating agent for Cd, Tl activation was much higher than that for Pb, especially Tl.

In practical applications, in order to avoid chelating agents poisoning plants, the activation effect under the condition of safe dosage is a problem worthy of attention. Taking common tolerant plant maize as an example, it has been found that the safe dosage of chelating agent (such as EDTA) is generally less than 3 mmol/kg [52,53]. Using 2.5 mmol/kg as the safe dosage, the maximum activation effect of various chelating agents was further calculated. Under the safe dosage, for Pb-contaminated soil, the six chelating agents all achieve the MEAE, EDTA (90.91%) > NTA (89.62%) > DTPA (89.13%) > GLDA (87.13%) > EDDS (76.38%) > CA (50.03%). For Cd-contaminated soil, only EDTA can achieve the MEAE (86.96%, when the EDTA dosage was 1.8 mmol/kg), the activation effect of GLDA, DTPA, NTA can reach 69.31%, 69.08% and 63.95%, respectively, equivalent to 82.29%, 80.61% and 77.22% of the MEAE. The efficiency of the remaining three chelating agents CA and EDDS were 15.17% and 11.48%, respectively, under the dosage of 2.5 mmol/kg. For Tl-contaminated soil, the activation efficiencies of six chelating agents under safe dosage conditions were lower than 3%. It can be seen that under the condition of safe dosage, the first type of chelating agent (APCAs including EDTA, NTA, DTPA, GLDA and EDDS) has a good activation efficiency in Pb-contaminated soil and all above 76%. For Cd-contaminated soil, the activation efficiency of EDTA, GLDA, DTPA and NTA can also be more than 63%. Furthermore, considering the degradation characteristics of chelating agent and the risk of groundwater pollution, DTPA and EDTA are difficult to degrade, so GLDA and NTA are the best choice for phytoremediation of both Pb- and Cd-contaminated red soils.

In addition, even if EDTA is non-biodegradable, the Pb and Cd chelate formed by it has a limited migration distance under local red soil conditions and does not affect groundwater [54]. Therefore, it is considered that the application of an appropriate dosage of chelating agent, even EDTA, will not pose a threat to groundwater. For Tl-contaminated soil, the activation efficiencies of the six chelating agents are very low under red soil and safe dosage, and the method of enhancing the activation efficiency needs further study.

On the basis of the above experimental results, it showed that the six chelating agents had a better activation effect on Pb- and Cd-contaminated soils under neutral pH conditions. Under acidic conditions, H^+^ will combine with chelating agents to form conjugate acids, thereby weakening the chelating reaction. Moreover, pH affects the stability of chelates between chelating agents and heavy metals, and it has been found that the stability constants of chelates formed between Pb-/Cd- and GLDA, EDTA, EDDS, IDSA, HIDS, MGDA are higher at pH = 6.5 than that at pH = 4 [55,56]. For the Tl-contaminated soil, the activation effects of the six chelating agents were slightly better under acidic conditions than in a neutral environment, although the maximum difference in activation efficiency was no more than 10%. This difference may be related to the enhancement of the cation alternating effect of heavy metal ions with H^+^ under acidic conditions [28].

## 5. Conclusions

Based on the experiments of chelating agent activation and speciation analysis of heavy metals in red soil, the following conclusions were obtained.

(1) The activation of Pb, Cd and Tl in red soil by chelating agent reached equilibrium within 2 h, and the activation effect showed the dynamic characteristics of three stages (fast, slow and equilibrium), which was related to non-specific and specific adsorption in red soil.

(2) Under neutral conditions, the equilibrium activation effect (EAE) of the six chelating agents (EDTA, GLDA, DTPA, NTA, EDDS and CA) approximately obeyed the law of logarithmic change with the increase of the chelating agents dosage, coefficients of determination ranging from 0.693 to 0.984. Compared with Tl-contaminated soil, chelating agents had better activation performance on Pb- and Cd-contaminated soils. Except for EDDS, CA, the maximum equilibrium activation effect (MEAE) of other chelating agents (EDTA, GLDA, DTPA and NTA) was over 81%, and Pb-contaminated soil was slightly better, which is closely related to the existing forms of Pb and Cd, and most of them exist in exchangeable and reducible fractions. Tl activation efficiency is very low (less than 15%), which is mainly related to the existential speciation of Tl (dominated by residual fraction), the stability constant of complexation reaction and the activation of non-target element Fe in red soil. Secondly, the order of the dosage of chelating agents corresponding to MEAE for three types of contaminated soils was Pb-contaminated soil > Cd-contaminated soil > Tl-contaminated soil.

Under acidic pH conditions, the activation efficiencies of chelating agents decreased in different degrees in Pb- and Cd-contaminated soils, whereas the activation efficiencies of chelating agents in Tl-contaminated soils were slightly enhanced.

(3) Taking maize as an example, considering the safe use of chelating agents to plants and the limited migration of chelated heavy metals in red soil, EDTA, GLDA, DTPA and NTA can be used as the best chelating agents for phytoremediation in red soil polluted by both Pb and Cd. When degradation is considered, GLDA and NTA are the best option.

## Figures and Tables

**Figure 1 ijerph-17-00497-f001:**
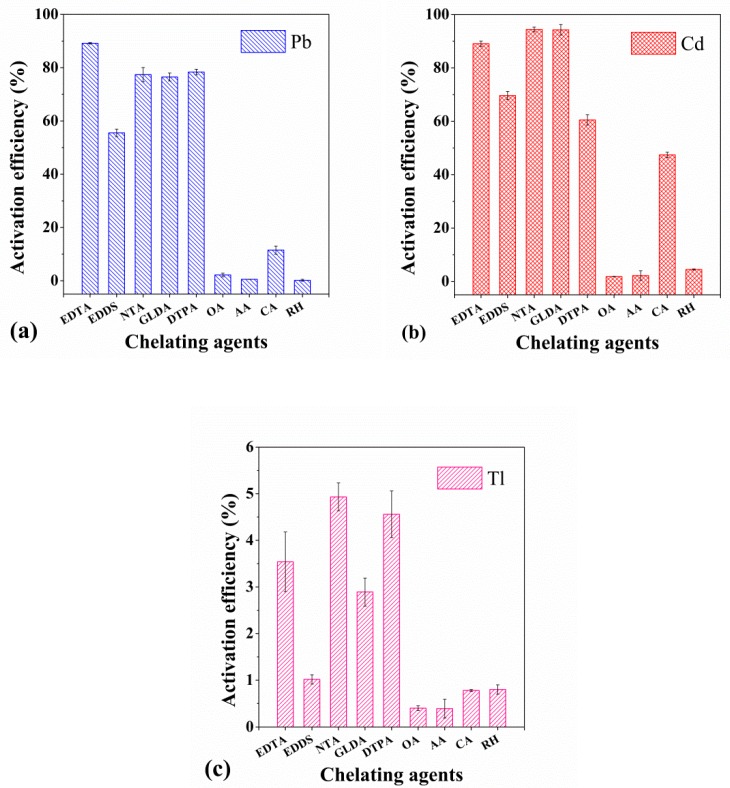
Maximum activation efficiency of Pb (**a**), Cd (**b**) and Tl (**c**) by different chelating agents. Error bars indicate the standard deviation (n = 3).

**Figure 2 ijerph-17-00497-f002:**
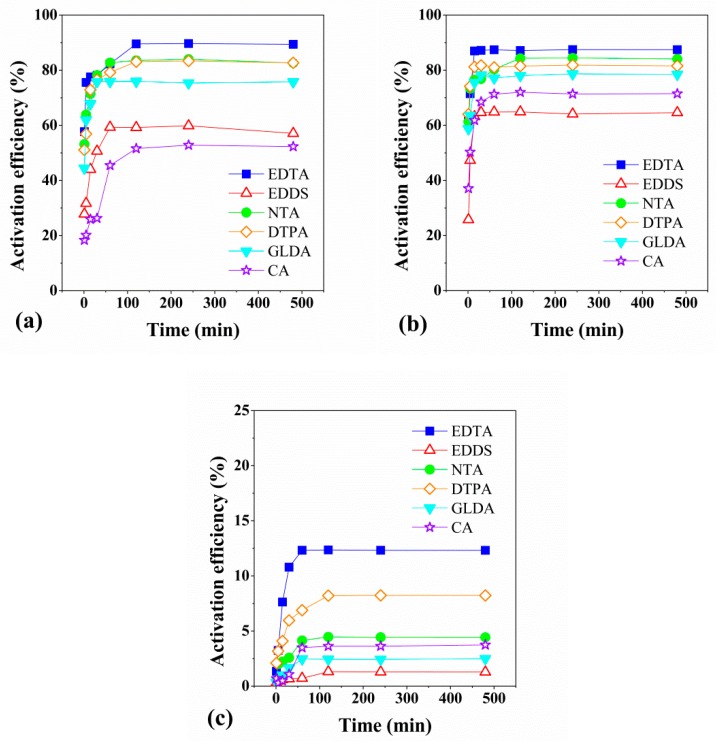
Activation kinetic curves of Pb (**a**), Cd (**b**), and Tl (**c**) with different chelating agents.

**Figure 3 ijerph-17-00497-f003:**
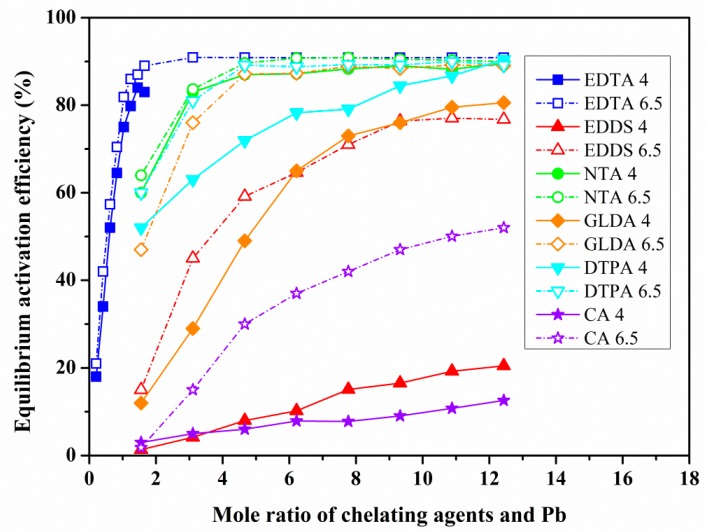
Effects of the dosage of different chelating agents on the activation of Pb at pH 6.5 and 4.

**Figure 4 ijerph-17-00497-f004:**
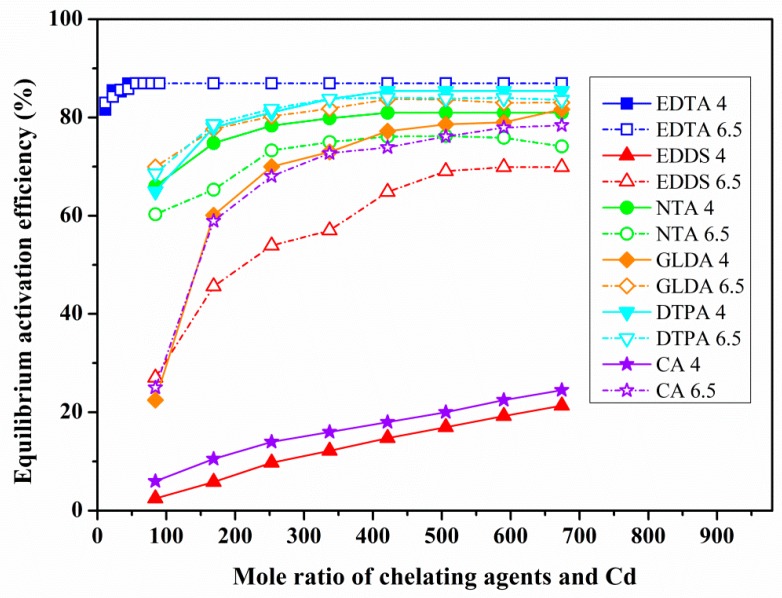
Effects of the dosage of different chelating agents on the activation of Cd at pH 6.5 and 4.

**Figure 5 ijerph-17-00497-f005:**
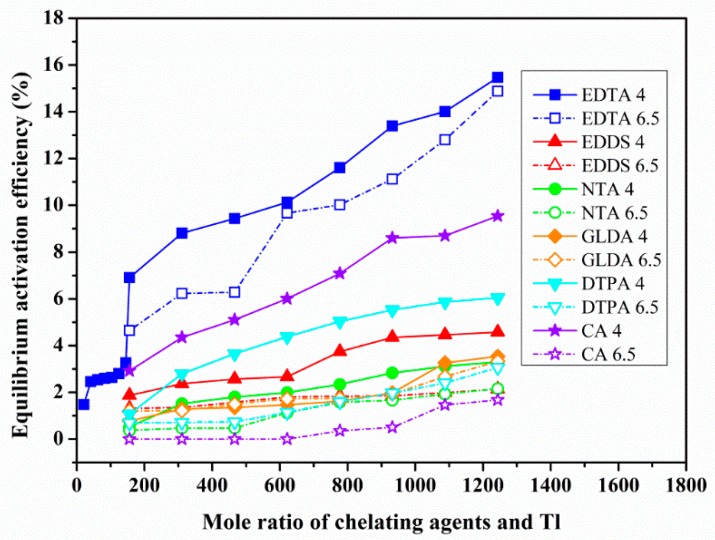
Effects of the dosage of different chelating agents on the activation of Tl at pH 6.5 and 4.

**Figure 6 ijerph-17-00497-f006:**
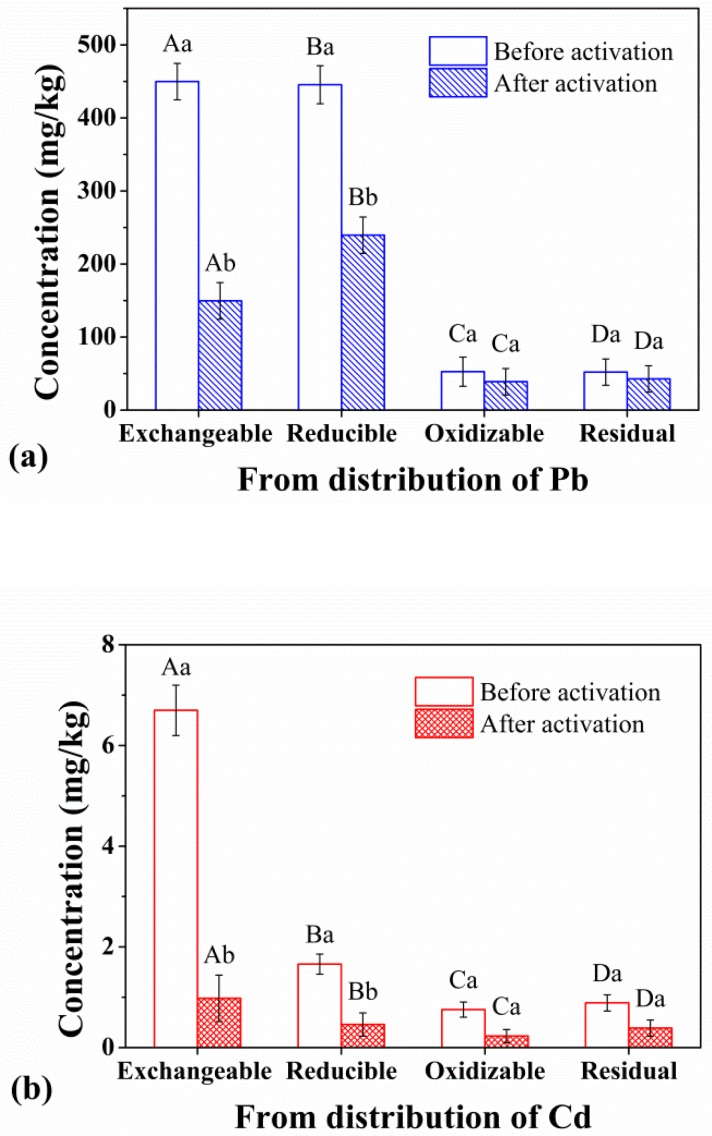
Speciation analysis of Pb (**a**), Cd (**b**), and Tl (**c**). Means of results from three parallel samples are presented, error bars represent standard deviations. A, B, C, D represent exchangeable, reducible, oxidizable, and residual fractions, respectively. Different lower-case letters indicate significant difference at *p* < 0.05 among different amounts of heavy metal under chelating agent-free/chelating agent-treated conditions, respectively.

**Table 1 ijerph-17-00497-t001:** Physiochemical properties of red soil.

Soil Properties		Value
Particle size	<0.002 mm (%)	16.00
	0.002–0.02 mm (%)	33.00
	0.02–2 mm (%)	53.00
pH		4.5
Available N (mg/kg)		37.80
Available P (mg/kg)		0.90
Available K (mg/kg)		12.10
SOM (g/kg)		15.40
CEC (cmol/kg)		3.30
Total heavy metal concentration (mg/kg)		
Pb		52.30
Cd		0.12
Tl		0.63

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
