# Peer review of "Comparative Activation Process of Pb, Cd and Tl Using Chelating Agents from Contaminated Red Soils"

_ijerph, 2020, doi:10.3390/ijerph17020497_

Round 1
Reviewer 1 Report
Suggestions in the attached file.

Reviewer 2 Report
This manuscript is interesting and generally easy to understand. It provides new and useful knowledge on the possibilities of forming soluble complexes between toxic elements present in contaminated red soils and complexing organic agents. The summary, however, is difficult to understand due to the use of the term "activation performance". The authors should rather speak in terms of the formation of soluble metal complexes which will be more readily available for extraction by phytoextraction. I have only one problem with the results presented. The authors make a direct link between the formation of soluble metal complexes and the potential of metals to be absorbed by plants by phytoextraction. However, the authors do not sufficiently discuss the importance of the organic complexing agent and its role in the phytoextraction process. Will all of the complexes formed allow the same yield for the uptake of metals by plants?Author Response
Please see the attachment.
